workplace; mental health; intervention; health promotion; health professionals; Africa

**Corresponding authors:**
Bala Isa Harri and Ejemai Eboreime;
Emails: bharri@dal.ca; ejemai.eboreime@dal.ca

# Addressing the mental health needs of healthcare professionals in Africa: a scoping review of workplace interventions

Bala Isa Harri[1,2], Igbekele Ogunboye[3], Adaeze Okonkwo[2], Aminu Yakubu[2], Janice Y. Kung[4], Jenson Fofah[5,6], Ojo Tunde Masseyferguson[7] and Ejemai Eboreime[1,8]

[1]Department of Psychiatry Dalhousie University, Halifax, NS, Canada; [2]Department of Health Planning Research and Statistics, Federal Ministry of Health, Abuja, Nigeria; [3]Department of Epidemiology and Biostatistics, School of Public Health, Jackson State University, Jackson, MS, USA; [4]Geoffrey & Robyn Sperber Health Sciences Library, University of Alberta, Edmonton, AB, Canada; [5]Royal United Hospital Bath, Bath, UK; [6]NHS, UK; [7]National Mental Health Programme, Department of Public Health, Federal Ministry of Health, Abuja, Nigeria and [8]Department of Psychiatry, University of Alberta, Edmonton, AB, Canada

## Abstract

Healthcare workers in Africa face considerable stress due to factors like long working hours, heavy workloads and limited resources, leading to psychological distress. Generally, countries in the global north have well-established policies and employee wellness programs for mental health compared to countries in the global south. This scoping review aimed to synthesize evidence from published and grey literature on workplace mental health promotion interventions targeting African healthcare workers using Social Ecological Model (SEM) and the Job Demands-Resources (JD-R) model as an underlying theoretical framework for analysis. Arksey and O'Malley framework for scoping reviews was used. The search was conducted across multiple databases. A total of 5590 results were retrieved from Ovid MEDLINE, Ovid Embase, Ovid PsycINFO, Cochrane Library, CINAHL, Scopus and Web of Science. Seventeen (17) studies from ten (10) African countries were included after title, abstract and full text screening. Thematic analysis identified 5 key themes namely training programs, counselling services, peer support programs, relaxation techniques and informational resources. In conclusion, even though limited workplace mental health interventions for healthcare professionals were identified in Africa, individual-level interventions have been notably substantial in comparison to organizational and policy-level initiatives. Moving forward, a multi-faceted approach unique to the African context is essential.

## Impact statements

This study provides a comprehensive review of workplace mental health promotion interventions for healthcare professionals across Africa. It reveals promising approaches and significant gaps in current research, policy and practice while offering valuable insights that could promote the development of a resilient health workforce through Individual, organizational and policy-level mental health interventions.

## Introduction

Health promotion extends beyond individual behaviour change by incorporating social and environmental interventions (World Health Organization, 2024a). The Ottawa Charter emphasizes the implementation of health promotion strategies in various community settings, including workplaces, prisons and schools (World Health Organization, 2024d). The workplace, in particular, offers a unique environment for health promotion due to the substantial amount of time adults spend at work and the diverse range of activities that take place there which can impact overall well-being. Consequently, the Luxembourg Declaration promotes a collaborative effort among employers, employees and society to enhance health and well-being in the workplace through Workplace Health Promotion (WHP) (European Network of Workplace Health Promotion, 2018). WHP involves creating a supportive work environment that promotes healthy behaviours, addresses health risks and enhances physical and mental health and well-being (Centers for Disease Control and Prevention, 2024b). While physical health workplace health promotion programs have been more prevalent, recent evidence highlights the importance of mental health workplace health promotion interventions and their impact on employee well-being and organizational performance (Søvold et al., 2021; World Health Organization, 2024c).

Mental health workplace health promotion focuses on implementing policies, programs and interventions to foster a supportive work environment that enhances employees' mental well-being. This includes awareness campaigns, stress management programs, work-life balance initiatives and access to mental health services (Wu et al., 2021; Centers for Disease Control and Prevention, 2024a). However, there are disparities in mental health workplace health promotion efforts across countries. Generally, countries in the global North exhibit higher awareness and recognition of mental health issues in the workplace, along with well-established policies, employee wellness programs, mental health training and available support resources. In contrast, countries in the global South face challenges such as limited resources, inadequate infrastructure, cultural barriers, lower awareness, stigma and limited access to mental health services (World Health Organization, 2024b).

The workplace of healthcare workers in Africa is characterized by considerable levels of stress. This stress primarily emanates from factors such as extended working hours, heavy workloads and limited resources (Dubale et al., 2019). Consequently, these stressors significantly contribute to the escalation of psychological distress and burnout among healthcare practitioners (Okwaraji and Aguwa, 2014; Søvold et al., 2021). Furthermore, the absence of adequate organizational support and resources specifically allocated to managing mental health exacerbates the aforementioned challenges (Søvold et al., 2021; Dawood et al., 2022). In this regard, healthcare workers are confronted with obstacles that impede their access to care, as the prevailing stigma and discrimination surrounding mental health discourage them from seeking assistance or openly acknowledging their personal struggles (Kapungwe et al., 2010; Egbe et al., 2014).

Furthermore, healthcare professionals in Africa frequently encounter traumatic experiences within their work environment, including infectious disease outbreaks and humanitarian crises. The exposure to such events heightens the risk of developing post-traumatic stress disorder (PTSD) and other related mental health conditions (De Boer et al., 2011; Greenberg et al., 2015). Complicating matters further, the prevailing work-life imbalance and the lack of emphasis on self-care practices serve to amplify these challenges faced by healthcare practitioners (Steele, 2020).

To ensure the well-being of healthcare workers in Africa, it is crucial to investigate how to address these pressing mental health concerns through comprehensive interventions that prioritize prevention, early intervention and accessible mental health support services within the workplace. Implementing robust interventions that target the mental well-being of healthcare professionals can foster a resilient and sustainable healthcare workforce, capable of providing optimal care to the population of Africa.

This scoping review aims to identify and synthesize evidence from published and grey literature on mental health promotion interventions designed for African healthcare professionals at their workplaces, encompassing all forms of research and policy documents. Furthermore, we seek to categorize these interventions based on their type, level of implementation (individual, organizational, or policy) and targeted outcomes, thereby providing a comprehensive overview of the current landscape and identifying gaps in research and practice.

## Methods

The study adopted the framework outlined by Arksey and Malley (2005) for conducting a scoping review (Arksey and O'Malley, 2005) This scoping review was reported according to the Preferred Reporting Items for Systematic Reviews and Meta-Analyses extension for Scoping Reviews (PRISMA-ScR).

### Theoretical framework

This scoping review is grounded in two interconnected theoretical perspectives: the Social Ecological Model (SEM) (Bronfenbrenner, 1979; Mcleroy et al., 1988) and the Job Demands-Resources (JD-R) model (Demerouti et al., 2001; Bakker and Demerouti, 2007). The SEM provides an overarching framework, positing that health behaviours and outcomes are influenced by multiple interacting levels: individual, interpersonal, organizational, community and policy. This multi-level perspective aligns with our thematic analysis, which identified interventions targeting various ecological levels. Within this broader ecological structure, the JD-R model offers insight into the specific mechanisms of workplace mental health, proposing that employee wellbeing is determined by the balance between job demands (aspects requiring sustained effort) and job resources (aspects that help achieve goals or reduce demands). Together, these frameworks guide our analysis of workplace mental health promotion interventions for healthcare professionals in Africa, helping to interpret results and inform discussions on practice and research implications. This integrated approach underscores the need for multi-level interventions that address both environmental factors and individual coping strategies, particularly in the resource-constrained and high-demand context of African healthcare settings.

### Inclusion and exclusion criteria

The study included all articles with primary, secondary and tertiary preventive interventions that promote the mental health of health workers at their workplace in Africa that are written or translated into English. Date of publication, quality of articles and methodology were not considered in the selection. Health promotion interventions that focused only on physical health of health workers at their workplace were excluded. Additionally, studies with interventions to address mental health of health workers in their workplace outside the African continent, and those not available or translated in English were excluded.

### Search strategy

Database searches were completed in Ovid MEDLINE, Ovid Embase, Ovid PsycInfo, Cochrane Library (via Wiley), CINAHL, Scopus and Web of Science core collection on the 4th August 2023 to retrieve all relevant literature pertaining to the mental health promotion interventions for healthcare professionals in Africa, relevant keywords, word phrases and controlled vocabulary were carefully selected. Boolean operators (AND, OR) were used in each of the databases to combine keywords, and their alternatives with applied wild cards or truncation to search for relevant studies. No language or date limits were applied. Studies were exported to a web-based tool called Covidence (www.covidence.org). Bibliographies from included studies were also reviewed and grey literature was searched on United Nations (UN) agency's websites such as the World Health Organization (WHO), International Labour Organization (ILO) and United Nations Development Programme (UNDP). To gather additional information, non-governmental organizations (NGOs) websites such as the African Centres for Disease Control and Prevention, Africa Mental Health Foundation (AMHF), African Mental Health Research Initiative (AMARI), Strong Minds and Basic

Needs Africa were searched. Supplementary Material, Appendix 1 shows full search strategies. All articles from the 7 databases were combined in Covidence and duplicates were removed. The title and abstract were then screened on Covidence by two researchers independently to exclude those that did not meet the inclusion criteria. Where there was disagreement, a third reviewer served as an arbitrator to reach a consensus. After the title and abstract screening, a full-text screening to exclude those that do not meet the criteria was done by two reviewers with a third reviewer involved in resolving disagreements.

### Data extraction

Microsoft Excel 365 was used for data extraction and analysis. The collated information includes. Study Title, Authors, Year of Publication, Country, Aim of studies, Workplace of Health Worker, Setting of Intervention, Sample Size, Study Design, Age Range of participants, Category of Healthcare Professionals, Type of intervention, Name of Mental Health Intervention, Description of Mental Health Intervention, Duration of Intervention, Frequency of intervention, Outcomes Measured, Key Findings.

### Data synthesis

A narrative approach was utilized to gather identified data. The data were generated based on countries within the African region, specific populations of interest (healthcare workers), the level of healthcare facility as their workplace and the mental health interventions provided. We employed tables to summarize the features of the studies and interventions, and a map to describe the countries where the studies were conducted.

### Results

A total of 5590 results were retrieved from databases. No relevant article was found from reference list searching and other websites. A total of 2,925 duplicates were removed and 2665 studies were screened against title and abstract. A total of 2528 studies were excluded after title and abstract while 137 studies assessed for full-text eligibility.

After full-text screening, 120 studies were excluded and 17 studies met the inclusion criteria. Figure 1 is the PRISMA Diagram while Tables 1 and 2 are the summary characteristics of included studies.

### Countries of study

Seventeen studies that met the inclusion criteria are from 10 African countries, namely Botswana, Egypt, Kenya, Malawi, Nigeria, Rwanda, Sierra Leone, South Africa, Tunisia and Zimbabwe. Figure 2 shows the map of countries included in the studies with intervention.

### Thematic analysis of included studies

This review identified six key themes highlighting workplace mental health interventions implemented for healthcare professionals in African. These themes reflect the diverse approaches taken to address the mental health needs of healthcare providers across various African contexts.

### Training programs

Training programs emerged as the most frequently implemented intervention type (n=8 studies), spanning multiple countries including Egypt, Kenya, Nigeria, Rwanda, Sierra Leone, South Africa, Tunisia and Zimbabwe. These programs focused on building critical skills such as stress management (Maingi et al., 2022; Zingela et al., 2022; Kleinau et al., 2024), assertiveness (Abdelaziz et al., 2020), psychological first aid (Maingi et al., 2022) and self-care(Vesel et al., 2015; Şahin et al., 2021) .

The majority of studies reported consistent short-term improvements in coping abilities, resilience, wellbeing and lower perceived stress following these training programs (Vesel et al., 2015; Waterman et al., 2018; Kelly et al., 2021; Şahin et al., 2021; Akinsulore et al., 2022). For instance, Zingela et al. (2022) found that their psychological preparedness training in South Africa enhanced healthcare workers' ability to manage outbreak-related stress and improved coping skills. Similarly, Vesel et al. (2015) reported improvements in communication, self-care and social connectedness among healthcare workers in Sierra Leone following their intervention.

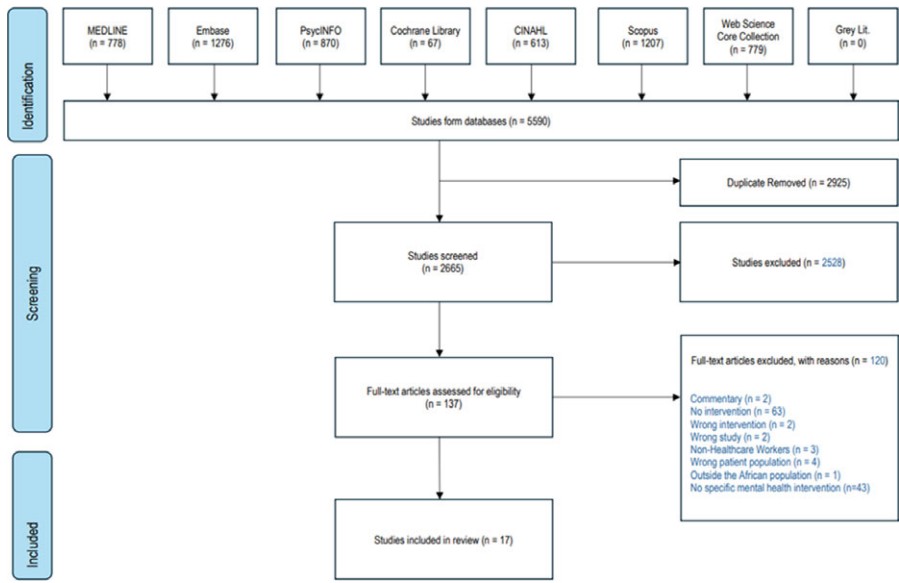

**Figure 1.** PRISMA diagram.

**Table 1.** Summary characteristics of included studies

| S/N | Study Title | Authors | Year of publication | Aim of studies | Study design | Sample size | Category of healthcare professionals (study participants) | Description of mental health status of participants at enrolment | Description of mental health intervention |
|---|---|---|---|---|---|---|---|---|---|
| 1 | Developing an mHealth Intervention to Reduce COVID–19– Associated Psychological Distress Among Health Care Workers in Nigeria: Protocol for a Design and Feasibility Study | A. Akinsulore, O. Aloba, O. Oginni, I. Oloniniyi, O. Ibigbami, C. Seun-Fadipe, T. Opakunle, A. Owojuyigbe, O. Olibamoyo, B. Mapayi, V. and Okorie, A. Adewuy (Akinsulore et al., 2022) | 04-Jan–22 | The overall aim of the study is to investigate COVID–19-associated psychosocial distress and evaluate the feasibility of using the m-Health-based intervention in managing this distress among health care workers in Nigeria. | Mixed Methods (Quantitative and Qualitative) | Quantitative Study: 440 nurses and doctors (healthcare workers) Qualitative Study: 60 in-depth interviews, 20 key informant interviews and 4 focus group discussions (24 participants) Mixed Methods: 40 participants in in-depth interviews | Doctors and Nurses | No description mental health status of participants' enrolment | The intervention mHealth encompasses using mobile devices to collect, store, retrieve and share information among users of the mHealth platform. Described as the 'therapist in the pocket' intervention, the m-Health is a treatment technique administered independently or as a supplement to extensively transform psychological treatment. The delivery of healthcare and public health services through mHealth relies heavily on using short message services (SMS), voice and multimedia services (MMS) on mobile phones. |
| 2 | Laughter therapy as an intervention to promote psychological well-being of volunteer community care workers working with HIV-affected families | Irene Hatzipapas, Maretha J. Visser, and Estie Janse van Rensburgc (Hatzipapas et al., 2017) | 14-Dec–17 | The research had two main objectives: firstly, to gain insight into the emotional experiences of community care workers who provide care for families affected by HIV, and secondly, to investigate the benefits of laughter therapy sessions as a form of self-care for these workers. The aim was to determine the value of such sessions within this context. | Mixed methods (quantitative and qualitative research technique) | 30 care workers (initial sample size); 10 participants (willing to do interviews): 3 male and 7 females; 7 participants (completed the interviews): 2 males and 5 females | Volunteer Community Health Workers | No description of mental health status participants at enrolment | Aerobic Laughter Therapy (ALT) is a complementary treatment that aids in coping with various challenges by encouraging playfulness and stress relief. Sessions begin with warm-up activities like stretching and clapping to promote childlike playfulness, followed by guided breathing and laughter exercises that combine acting and visualization techniques. |
| 3 | The development of a model for dealing with secondary traumatic stress in mental health workers in Rwanda | Jean Damascene Iyamuremye, and Petra Brysiewicz (Iyamuremye and Brysiewicz, 2015) | 24-Feb–15 | The aim is to develop a comprehensive model that effectively manages the effects of secondary traumatic stress (STS) on mental health workers in Rwanda. This model will integrate primary, secondary, and tertiary interventions to | Quantitative design and Qualitative design using Collaborative Action Research Approach | 180 participants for the Quantitative design; 30 mental health workers for the Qualitative Design (unstructured interviews) | Mental health workers (nurses, doctors, psychologists, trauma counsellors and social workers | Participants were above the cut-off levels for significant traumatisation and at risk of secondary traumatic stress (73.8% personally experience the 1994 genocide 10% experienced accidental disaster | The Intervention Model to Manage Secondary Traumatic Stress (IMMSTS) was developed for mental health workers in Rwanda, addressing the high levels of secondary traumatic stress experienced by professionals who were often themselves victims of the 1994 genocide. It |

| S/N | Study Title | Authors | Year of publication | Aim of studies | Study design | Sample size | Category of healthcare professionals (study participants) | Description of mental health status of participants at enrolment | Description of mental health intervention |
|---|---|---|---|---|---|---|---|---|---|
| | | | | manage the impact of STS on mental health workers in Rwanda. | | | | 7.7% had experienced emotional and psychological abuse 7.2% had experienced some kind of natural disaster 2.2% physical abuse as a child) | comprises three main components: prevention (including education, self-awareness and calmness techniques), assessment (both individual and organizational risk evaluation), and treatment (self-care strategies and therapeutic approaches). |
| 4 | Psychosocial support and resilience building among health workers in Sierra Leone: Interrelations between coping skills, stress levels and interpersonal relationships. | Linda Vesel1, Kathryn Waller, Justine Dowden, Jean Christophe Fotso (Vesel et al., 2015) | 8-Jun–15 | The specific aims were to improve coping techniques among health workers by addressing workplace stressors and introducing support services and to improve interpersonal relationships between health workers and with client | Mixed Methods | 271 | Community Health Officers, Maternal and Child Health Aides, Registered Nurses, Vaccinators, Nursing Aides, Community Health Nurses and Endemic Disease Control Unit Assistants; | No description of mental health status participants at enrolment | The Helping Health Workers Cope (HHWC) intervention includes training on communication, self-care and social connectedness to help health workers better manage work-related stress and improve their relationships with colleagues and clients |
| 5 | Training peers to treat Ebola centre workers with anxiety and depression in Sierra Leone. | Waterman, Samantha; Hunter, Elaine Catherine Margaret; Cole, Charles L; Evans, Lauren Jayne; Greenberg, Neil; Rubin, G James; Beck, Alison (Waterman et al., 2018) | 10-Jul–05 | To train Ebola Treatment Centre (ETC) staff to provide a 3-phase CBT based intervention for common mental health problems to fellow ETC staff and explored the effectiveness of the intervention | Pre and post intervention assessment(quasi-experiment) | 3273 | Not specified | No description of mental health status participants at enrolment | The intervention is a phased, CBT-based group program provided to staff at an Ebola Treatment Center (ETC) in Sierra Leone after the Ebola crisis. It included three key phases: A 2-hour Psychological First Aid workshop where staff could discuss work-related challenges and coping strategies. Targeted workshops addressing specific mental health issues identified during the initial screening. Intensive CBT-based interventions delivered by trained local facilitators, with remote support from UK clinicians. |

| S/N | Study Title | Authors | Year of publication | Aim of studies | Study design | Sample size | Category of healthcare professionals (study participants) | Description of mental health status of participants at enrolment | Description of mental health intervention |
|---|---|---|---|---|---|---|---|---|---|
| 6 | Developing a healthcare worker psychological preparedness support programme for the COVID–19 outbreak | Zukiswa Zingela Stephan van Wyk Aletta Bronkhorst Carmenita Groves (Zingela et al., 2022) | 10-Mar–22 | To develop a psychological preparedness training (PPT) programme to support frontline health workers | Observational, descriptive and cross-sectional design with pre- and post intervention analysis | 761 | Not specified | No description of mental health status participants at enrolment | The intervention is a group psychological preparedness training (PPT) developed and implemented to support healthcare workers during the COVID–19 outbreak. It focused on helping healthcare workers identify and manage thoughts, feelings and behaviours related to the outbreak, develop coping strategies and reinforce team strengths. |
| 7 | Effectiveness of a chatbot in improving the mental wellbeing of health workers in Malawi during the COVID–19 pandemic: A randomized, controlled trial | Eckhard F. Kleinau1*, Tilinao Lamba2, Wanda Jaskiewicz3, Katy Gorentz3, Ines Hungerbuehler4, Donya 6 Rahimi3, Demoubly Kokota2, Limbika Maliwichi2, Edister S. Jamu2, Alex Zumazuma5, Mariana Negrão4, 7 Raphael Mota4, Yasmine Khouri4, Michael Kapps4 (Kleinau et al., 2024) | 28-Jan–2023 | To investigate the working hypothesis that a virtual mental healthcare assistant chatbot, Vitalk, is an acceptable source of psychosocial and mental wellbeing support for health workers to effectively decrease work-related anxiety, depression, burnout and loneliness (based on standard mental health scales) and to increase resilience and resilience-building behaviors. | RCT | 1584 enrolled, 836 completed | Doctors Nurses Medical Assistants Clinical officers Laboratory technicians Physiotherapy technicians Pharmacists Physiotherapists | About 1 in 8 participants (approximately 12–13%) reported anxiety and depression - Mean baseline anxiety score (GAD–7) was 4.5 - Mean baseline depression score (PHQ–9) was 4.0 2. Burnout: Approximately 3 in 4 participants (75%) suffered from burnout - Mean baseline burnout score (OLBI) was 38.4 3. Resilience: - About 1 in 4 participants (25%) had low resilience levels - Mean baseline resilience score (RS–14) was 78.6, indicating moderate to moderately high resilience - Mean baseline resilience-building activities score was 13.1, indicating | Vitalk is an automated mental health chatbot app that delivers content through interactive conversations based on Cognitive Behavioral Therapy (CBT) and Positive Psychology. The control group accessed four static mental health resources through internet links to read webpages |

| S/N | Study Title | Authors | Year of publication | Aim of studies | Study design | Sample size | Category of healthcare professionals (study participants) | Description of mental health status of participants at enrolment | Description of mental health intervention |
|---|---|---|---|---|---|---|---|---|---|
| | | | | | | | | moderate level 4. Loneliness: - Mean baseline UCLA loneliness score was 5.3 | |
| 8 | Implementation of a National Workplace Wellness Program for Health Workers in Botswana | Jenny H. Ledikwe, PhD, Bazghina-werq Semo, MD, Miram Sebego, PhD, Maureen Mpho, BSc, Heather Mothibedi, MSc, Shreshth Mawandia, MSW, MPH and Gabrielle O'Malley, PhD (Ledikwe et al., 2017) | 24-Jul–2017 | The aim of the study was to assess the level of implementation of the National Workplace. Wellness Program (WWP) in Botswana and identify barriers to and facilitators of implementation. The WWP aimed to empower health workers with knowledge and skills to manage and cope with the dynamic demands of the health care system, which had been exacerbated by the HIV/AIDS epidemic. | Sequential, explanatory, mixed methods design including a national implementation assessment and in-depth interviews | National implementation assessment of 27 Health districts and and 38 in-depth interviews | All workers in healthcare facilities including non health professionals | No description of mental health status participants at enrolment | The intervention involved the implementation of Botswana's National Workplace Wellness Program (WWP) for healthcare workers (HCWs) which included health screenings, treatment, and care, focusing on conditions like HIV, tuberculosis and cancers, along with health promotion activities such as seminars and health talks. Stress management and team-building workshops, occupational health and safety measures, psychosocial and spiritual careand therapeutic recreation |
| 9 | Mitigation of Mental Health Effects Covid–19 Pandemic among Healthcare Workers in Western Kenya | Maingi, Z.; Kathukumi, K.; Jaika, S.; Odera, P.; Konyole, S.; Tibbs, C.(Maingi et al., 2022) | 09-Jun–22 | To investigate the measures adopted by HCW in western Kenya to mitigate the mental health effect of the COVID–19 | Cross sectional Descriptive Study Design | 356 | Medical Doctors Clinical Officers Nurses Nutritionist Medical Laboratory Officers Social Workers Psychologist Counsellors Radiographer Pharmacist invol in COVID 19 care | No description of mental health status participants at enrolment | Government guideline on mental health and psychological support during COVID–19: Recommended Sufficient rest at work Eating health food Engaging in physical activity Staying connected with friends and family. Avoiding unhelpful coping strategies such as alcohol consumption and smoking) |
| 10 | Effects of music therapy on occupational stress and burn-out risk of operating room staff. | I. Kacem, M. Kahloul, S. El Arem, S. Ayachi, M. Hafsia, M. Maoua, M. Ben Othmane, O. El Maalel, W. Hmida, O. | 25-May–20 | It evaluates the effects of music therapy program on the level of stress and burnout risk among the operating room staff of urology and maxillofacial | Quasi-experimental study with pre-post measures | 34 participants | Surgeons, Anesthetist doctors, Anesthetist technicians, Nurses, Instrumentalists and Caregivers | Using the Perceived Stress Scale (PSS–10) and the Maslach Burnout Inventory (MBI). 41.2% of participants had high levels of | Three daily music therapy sessions, each lasting 30 minutes, were provided to the operating room staff during working days. If a patient-related event occurred, the session was |

| S/N | Study Title | Authors | Year of publication | Aim of studies | Study design | Sample size | Category of healthcare professionals (study participants) | Description of mental health status of participants at enrolment | Description of mental health intervention |
|---|---|---|---|---|---|---|---|---|---|
| | | Bouallague, K. Ben Abdessalem, W. Naija & N. Mrizek (Kacem et al., 2020) | | surgery in an academic hospital. | | | | perceived stress. Burnout Indicators: Emotional Exhaustion: 38.2% of participants had high scores. Depersonalization: 50% had high scores. Professional Achievement: 58.8% had low scores. Burnout Syndrome: 17.2% of participants had a high level of burnout | rescheduled. A generic brand CD player was used in all operating rooms and the musical repertoire varied, including oriental, occidental and Tunisian music to accommodate participants' preferences. 4o |
| 11 | Does mindfulness reduce perceived stress in healthcare professionals? | Kckaou, A., Dhouib, F., Kotti, N., Sallemi, I., Hammami, K. J., Masmoudi, M. L., & Hajjaji, M. (Kckaou et al., 2023) | 18-Nov–22 | The aim is to explores the associations between mindfulness, perceived stress and well-being and life satisfaction across different professional categories. | Cross-sectional correlational study | 400 questionnaires distributed, 317 were returned, but only 297 questionnaires (74.25%) were included for data analysis. | Staff nurses, medical technicians and doctors | No description of mental health status participants at enrolment | The study explores the relationships between mindfulness and other factors like perceived stress, well-being and life satisfaction among healthcare professionals. |
| 12 | Improving Healthcare Worker Resilience and Well-Being During COVID–19 Using a Self-Directed E-Learning Intervention | Frances Kelly, Margot Uys, Dana Bezuidenhout, Sarah L. Mullane and Caitlin Bristol(Kelly et al., 2021) | 02-Dec–21 | The aim is to explores if there were any associations between behaviours, resilience and well-being. | Cross-sectional study | 474, participants that completed both the pre- and post-training assessments | Audiologist, speech therapist, Clinical Associate, Dentistry and Oral hygiene, Homoeopath, Medical Practitioner, Nursing, Occupational Therapist, Optometrist, Paramedic, Pharmacist, Physio, Chiro, Dietician, Biokineticist, Podiatry, Radiography, sonography, radiotherapist, Registered counsellor, psychologist and social worker | No description of mental health status participants at enrolment | A self-paced, online learning course designed to support healthcare worker well-being and resilience during the COVID–19 |
| 13 | Caring for the careers: A psychosocial | Idah Moyo, Livhuwani Tshivhase, & | 21-Jun–2023 | to develop a psychosocial support model that sustains a | Modelling | Not applicable | Nurses | No description of mental health status | The intervention is a psychosocial support model for healthcare |

(*Continued*)

| S/N | Study Title | Authors | Year of publication | Aim of studies | Study design | Sample size | Category of healthcare professionals (study participants) | Description of mental health status of participants at enrolment | Description of mental health intervention |
|---|---|---|---|---|---|---|---|---|---|
| | support model for healthcare workers during a pandemic | Azwihangwisi H. Mavhandu-Mudzusi (Moyo et al., 2023) | | support structure that will contribute to an enabling work environment promoting efficiency and effectiveness in response to public health emergencies | | | | participants at enrolment | workers during pandemics, including structured support with dedicated staff, counseling and follow-up for staff and families, a virtual crisis support, a welfare budget, training and follow-up for sick workers, effective communication, an anonymous platform for sharing experiences and addressing resource shortages with adequate protective equipment |
| 14 | How do family supportive supervisors affect nurses' thriving: Research before and during COVID–19 pandemic? | Şahin S, Adegbite WM, Tiryaki Şen H. (Şahin et al., 2021) | 24-Aug–2021 | To examine the effects of Covid–19 pandemic on nurses' perceived family supportive supervisor behaviors, work-to-family conflict, psychological well-being and thriving; and To test the effects of nurses' perceived family supportive supervisor behaviours on their thriving through work-to-family conflict and psychological well-being | Cross sectional | 511 | Nurses | No description of mental health status participants at enrolment! | The intervention uses family-supportive supervisor behaviours to enhance nurses' thriving, reduce work-to-family conflict and improve psychological well-being. These behaviours involve providing emotional and practical support, role modelling and implementing creative strategies to help nurses effectively manage their work and family responsibilities |
| 15 | The effectiveness of assertiveness training program on psychological wellbeing and work engagement among novice psychiatric nurses | Enas Mahrous Abdelaziz, Iman Abdelmotelb Diab, Marwa Mohamed Ahmed Ouda, Nadia Bassiouni Elsharkawy, Fadia Ahmed Abdelkader (Abdelaziz et al., 2020) | 07-Feb–2020 | To assess the effectiveness of an assertiveness training program on psychological wellbeing and work engagement among novice psychiatric nurses | A quasi-experimental design, single group (pre-post comparisonwithout a control group) | 36 | Nurses | No description of mental health status participants at enrolment | The study uses assertiveness training program for novice psychiatric nurses to enhance their assertiveness, psychological well-being and work engagement. Over seven weeks with twice-weekly sessions, the program covered assertive communication and anger management through lectures and role-playing. |

**Table 1.** (*Continued*)

| S/N | Study Title | Authors | Year of publication | Aim of studies | Study design | Sample size | Category of healthcare professionals (study participants) | Description of mental health status of participants at enrolment | Description of mental health intervention |
|---|---|---|---|---|---|---|---|---|---|
| 16 | Coping with COVID: Developing a Rapid-cycle Frontline Quality-improvement Process to Support Employee Well-being and Drive Institutional Responsiveness in a Tertiary Care Faith-based Hospital in Rural Kenya | Mary B. Adam, Naomi Wambui Makobu, Wilson Karuri Kamiru, Simon Mbugua and Faith Mailu (Adam et al., 2021) | 15-Jun–21 | To determine personal coping strategies used by staff and provide an opportunity for staff cross-learning; ask staff about what they need most; and provide a real-time feedback loop for decision-makers to support staff while coping with and managing stress during the COVID–19 outbreak | qualitative with a focus-on-focus group discussion | 122: in 17 focus group discussion sessions | All categories: Housekeeping, Housing, Kitchen, Nursing, Pharmacy, Laboratory, Radiology, Sewing, Nutrition, Physiotherapy, Finance, Clinical Officers and Medical Officer Interns | No description of mental health status participants at enrolment | The rapid cycle debrief session intervention consisted of conducting focus group discussions with frontline staff, excluding managers. The participants share their experiences and needs anonymously using sticky notes. Over a two-week period, a total of 17 focus groups were conducted, personal coping strategies, staff needs and provide a real-time feedback loop for management |
| 17 | Psychological support unit design and implementation during COVID–19 pandemic: Case of Mongi Slim Hospital, Tunisia | Wafa Abdelghaffar, Nadia Haloui, Noamen Bouchrika, Souha Yaakoubi, Amani Sarhane, Emna Kalai, Nihel Siala, Hajer Boulehmi, Souad Trabelsi, Soumaya Bourgou, Fatma Charfi, Ahlem Belhadj, Rym Rafrafi (Abdelghaffar et al., 2021) | 24-Jun–21 | To describe the design and implementation of a Psychological Support Unit for staff and patients of a hospital | Descriptive activities of a PSU during COVID 19 | Not specified | not specified | No description of mental health status participants at enrolment | The Psychological Support Unit (PSU) at established during the COVID–19 pandemic to offer mental health support. It provided prevention and care activities for patients, families and healthcare professionals, including a free helpline, stress management workshops, debriefing sessions and support groups. |

**Table 2.** Summary characteristics of included studies

| S/N | Authors | Outcomes measured | Tools used in measurement | Facility level of workplace | Level of prevention | Quantitative results | Key findings | SEM Policy Organizational (institutions) Interpersonal (families, friends, social networks) Individual (Knowledge, attitudes, skills) | Job demand -Resources |
|---|---|---|---|---|---|---|---|---|---|
| 1 | A. Akinsulore, O. Aloba, O. Oginni, I. Oloniniyi, O. Ibigbami, C. Seun-Fadipe, T. Opakunle, A. Owojuyigbe, OOlibamoyo, B. Mapayi, V. and Okorie, A. Adewuy (Akinsulore et al., 2022) | Psychological distress, changes in depressive and anxiety symptoms. Secondary outcomes focus on the feasibility, usability engagement, satisfaction, acceptability of mobile health (mHealth) interventions | 1. Kessler Psychological Distress Scale 2. The 9-item Patient Health Questionnaire 3. The 7-item Generalized Anxiety Disorder Scale 4. The Ssystem Usability Scale 5. The Mobile App Rating Scale | Tertiary | Secondary | No available quantitative results as the paper described the study still in progress | These findings underscore the urgent need for targeted psychological support for healthcare workers and highlight the potential of mHealth interventions to provide accessible mental health care. By offering support remotely, these interventions can overcome barriers of distance and offer the added advantage of privacy, helping to reduce the stigma often associated with seeking mental health support in traditional healthcare settings. | Individual level | |
| 2 | Irene Hatzipapas, Maretha J. Visser, and Estie Janse van Rensburgc (Hatzipapas et al., 2017) | Anxiety, depression, Perceived Stress and coping mechanisms | 1. Perceived stress scale (PSS) 2. Hospital anxiety and depression scale (HADS) | Primary | Secondary | Due to small sample size (7) no broad statistical claims made . | The study found that laughter therapy sessions had a beneficial impact on community care workers attending to HIV-affected families. The intervention reduced anxiety, depression and stress levels among participants, leading to more positive emotions, improved social relationships and better coping mechanisms. | Individual level | |
| 3 | Jean Damascene Iyamuremye, and Petra Brysiewicz (Iyamuremye and Brysiewicz, 2015) | Secondary traumatic stress (STS) on mental health workers and personal experiences, emotional impacts and coping mechanisms related to STS. | 1. Trauma Attachment Belief Scale (TABS) 2. Intervention Model to Manage Secondary Traumatic Stress (IMMSTS) | Community Care | Primary and secondary | Specific quantitative results after implementing the model are not provided The quantitative results pre-intervention: 73.8% experienced genocide events Mean TABS (Trauma Attachment Belief Scale) score was 77.0 (sd 1.2) (above the cut-off point of 50 for significant traumatization) | The model tend offer mental health professionals an effective framework for addressing the issue of STS | Individual / Organizational level | Job demand resources |

**Table 2.** (*Continued*)

| S/N | Authors | Outcomes measured | Tools used in measurement | Facility level of workplace | Level of prevention | Quantitative results | Key findings | SEM Policy Organizational (institutions) Interpersonal (families, friends, social networks) Individual (Knowledge, attitudes, skills) | Job demand -Resources |
|-----|---------|-------------------|---------------------------|----------------------------|--------------------|---------------------|--------------|-----------------------------------|-----------------------|
| 4 | Linda Vesel1, Kathryn Waller, Justine Dowden, Jean Christophe Fotso(Vesel et al., 2015) | Perceived stress levels, coping skills and interpersonal relationships. | 1. Proforma form | Primary | primary | Overall coping strategies increased from 2.63 (pre-test) to 3.23 (post-test) (p=0.000) Communication skills increased from 2.26 to 3.42 (p=0.000) Self-care skills increased from 2.72 to 2.99 (p=0.000) Social connectedness increased from 2.81 to 3.32 (p=0.000) Stress Levels: Post-intervention stress levels were lower in intervention district (2.40) compared to control district (2.48) (p=0.034) Relationships (measured on a 4-point scale): Overall relationships improved from 2.69 to 3.47 (p=0.000) Relationships with co-workers at facility improved from 2.67 to 3.55 (p=0.000) Relationships with patients improved from 2.66 to 3.42 (p=0.000) Relationships: Overall relationships: 3.47 vs 3.35 (difference +0.121, p=0.025) With co-workers at facility: 3.55 vs 3.44 (difference +0.111, p=0.110) - not statistically significant With patients: 3.42 vs 3.49 (difference −0.072, p=0.320) - not statistically significant | The study revealed notable improvements in coping skills, stress management and relationships with colleagues and clients in the intervention group compared to both pre-intervention and control groups. Health workers showed enhanced abilities in communication, self-care and social connectedness following the intervention. | Individual/ Interpersonal Level | |

*Cambridge Prisms: Global Mental Health*

| S/N | Authors | Outcomes measured | Tools used in measurement | Facility level of workplace | Level of prevention | Quantitative results | Key findings | SEM Policy Organizational (institutions) Interpersonal (families, friends, social networks) Individual (Knowledge, attitudes, skills) | Job demand -Resources |
|---|---|---|---|---|---|---|---|---|---|
| 5 | Waterman, Samantha; Hunter, Elaine Catherine Margaret; Cole, Charles L; Evans, Lauren Jayne; Greenberg, Neil; Rubin, G James; Beck, Alison(Waterman et al., 2018) | Stress, sleep, anxiety, depression, behavioural changes, relationship difficulties and post-traumatic stress disorder (PTSD). | 1. item wellbeing screening tool 2. Post-Traumatic Stress Checklist – Civilian version (PCL-C) 3. Perceived Stress Scale (PSS) 4. Insomnia Severity Index (ISI) 5. Generalised Anxiety Disorder 7 (GAD7) 6. Patient Health Questionnaire 9 (PHQ9) 7. Relationship Questionnaire 8. Behaviour questionnaire | Tertiary | Secondary | Phase 2 to phase 3 Changes: stress decreased from $27.77 \pm 7.63$ to $23.37 \pm 6.02$ ($p < 0.05$) Anxiety decreased from $16.88 \pm 3.83$ to $13.76 \pm 6.77$ ($p < 0.05$) Depression decreased from $22.10 \pm 4.31$ to $15.56 \pm 9.16$ ($p < .01$) Behavioral problems decreased from $1.30 \pm 1.34$ to $0.53 \pm 1.11$ ($p < .05$) Alcohol usage decreased from $3.69 \pm 4.53$ to $1.54 \pm 2.86$ ($p < .05$) Phase 3 pre-post changes: Wellbeing screening measure improved from $44.61 \pm 16.05$ to $33.93 \pm 15.75$ ($p < .01$) PTSD symptoms decreased from $59.39 \pm 17.86$ to $46.41 \pm 19.53$ ($p < .01$) Stress decreased from $23.58 \pm 5.50$ to $20.58 \pm 4.44$ ($p < .01$) Sleep problems decreased from $24.23 \pm 8.91$ to $19.60 \pm 7.63$ ($p < .01$) Anxiety decreased from $13.52 \pm 6.35$ to $10.40 \pm 6.48$ ($p < .05$) Depression decreased from $15.32 \pm 8.23$ to $12.60 \pm 7.70$ ($p < .05$) Anger decreased from $10.60 \pm 6.11$ to $7.43 \pm 5.87$ ($p < .01$) Relationship difficulties decreased from $27.61 \pm 5.87$ to $23.78 \pm 6.05$ ($p < .01$) | Effectively reduced mental health symptoms among health and Significant improvements were observed across multiple measures, including stress, depression, anxiety, behaviour and relationships | Individual level | |
| 6 | Zukiswa Zingela Stephan van Wyk Aletta Bronkhorst Carmenita Groves(Zingela et al., 2022) | Psychological preparedness, stress management and coping abilities | 26-item audit tool used to evaluate healthcare workers' knowledge, preparedness, coping ability and stress management related to the COVID–19 outbreak. | Tertiary | Primary | Statistical analysis results: 26-item audit tool Showed statistically significant improvement from pre- to post-intervention ($M = 2.77$, SD = 0.66 pre-intervention to $M = 3.57$, SD = 0.44 post-intervention; | The intervention enhanced healthcare workers' ability to manage outbreak-related stress, improved coping skills and fostered better teamwork and collaboration among staff. | Individual level | |

*(Continued)*

**Table 2.** (*Continued*)

| S/N | Authors | Outcomes measured | Tools used in measurement | Facility level of workplace | Level of prevention | Quantitative results | Key findings | SEM Policy Organizational (institutions) Interpersonal (families, friends, social networks) Individual (Knowledge, attitudes, skills) | Job demand -Resources |
|---|---|---|---|---|---|---|---|---|---|
| | | | | | | $t = -9.7$, df = 144.22, $p < 0.001$) For the 10-item tool: Showed statistically significant improvement from pre- to post-intervention ($M$ = 2.44, SD = 0.58 pre-intervention to $M$ = 3.11, SD = 0.70 post-intervention; $t = -10.87$, df = 159.77, $p < 0.001$) | | | |
| 7 | Eckhard F. Kleinau1*, Tilinao Lamba2, Wanda Jaskiewicz3, Katy Gorentz3, Ines Hungerbuehler4, Donya 6 Rahimi3, Demoubly Kokota2, Limbika Maliwichi2, Edister S. Jamu2, Alex Zumazuma5, Mariana Negrão4, 7 Raphael Mota4, Yasmine Khouri4, Michael Kapps4(Kleinau et al., 2024) | Depression Burnout Anxiety Loneliness Resilience Mood | 1. Depression-Patient Health Questionnaire (PHQ–9) 2. Burnout-Oldenburg Burnout Inventory (OLBI) 3. Anxiety -Generalized Anxiety Disorder (GAD–7) 4. Loneliness-UCLA Short (three-item) Loneliness 5. Scale (UCLA Loneliness) 5. Resilience–5-item Resilience-Building Behaviour Scale 6. 14-item Resilience Scale (RS–14) Mood-Mood meter | Primary, Secondary and Tertiary | Secondary | Difference-in-differences (DiD) estimators: Depression: −0.68 [95% CI −1.15 to −0.21] Anxiety: −0.44 [95% CI −0.88 to 0.01] Burnout: −0.58 [95% CI −1.32 to 0.15] Resilience: 1.47 [95% CI 0.05 to 2.88] Resilience-building activities: 1.22 [95% CI 0.56 to 1.87] Effect sizes (Cohen's $d$): Treatment group showed medium effect sizes: Depression: −0.41 Burnout: −0.36 Anxiety: −0.32 Resilience: 0.42 Resilience-building: 0.78 Reliable change percentages: treatment group: Depression: 17% improved Anxiety: 24% improved Resilience-building: 26% improved Resilience: 13% improved Mean scores at baseline vs endline: treatment Group: Anxiety: 4.73–2.93 Depression: 4.29–2.72 Burnout: 38.84–36.63 Resilience: 77.69–82. | Both Vitalk and access to a webpage intervention resulted to improvements in mental wellbeing and resilience. However, the effect size was consistently larger for the treatment group with Vitalk than for the control group with access to webpage. | Individual level | |

(*Continued*)

*Cambridge Prisms: Global Mental Health*

| S/N | Authors | Outcomes measured | Tools used in measurement | Facility level of workplace | Level of prevention | Quantitative results | Key findings | SEM Policy Organizational (institutions) Interpersonal (families, friends, social networks) Individual (Knowledge, attitudes, skills) | Job demand -Resources |
|---|---|---|---|---|---|---|---|---|---|
| 8 | Jenny H. Ledikwe, PhD, Bazghina-werq Semo, MD, Miram Sebego, PhD, Maureen Mpho, BSc, Heather Mothibedi, MSc, Shreshth Mawandia, MSW, MPH and Gabrielle O'Malley, PhD(Ledikwe et al., 2017) | level of implementation of the national Workplace Wellness Program | Implementation Assessment Summary score | Tertiary | Primary | There was one main quantitative measurement used | The National Workplace Wellness Program for health workers had varied in implementation across districts, with more focus on health screenings and promotion than on occupational health and psychosocial services. Success was driven by wellness committees, administrative support and cultural integration, but challenges included competing priorities, limited technical capacity, discomfort among workers and a lack of emphasis on personal wellness | Policy level | |
| 9 | Maingi, Z.; Kathukumi, K.; Jaika, S.; Odera, P.; Konyole, S.; Tibbs, C. (Maingi et al., 2022) | Measure to mitigate mental health effect of COVID–19/Prevalence of mental disorders | Semi structure questionnaire adapted from PHQ–9 | Secondary and Primary | Primary | Measures Adopted to Promote Mental Wellbeing: 43.3% ($n = 154$) engaged in physical activity 80.1% ($n = 285$) consumed healthy, sufficient diet 79.2% ($n = 282$) engaged in caring for others 79.2% ($n = 282$) kept active 79.5% ($n = 283$) maintained contact with family and friends 69.9% ($n = 249$) used social media 72.2% ($n = 257$) talked about their feelings 84.8% ($n = 302$) practiced acceptance of the situation | The most used coping strategies include acceptance of situation, healthy eating, maintain family and friendship ties with family being the most used strategies . | Individual/ Interpersonal level | |
| 10 | I. Kacem, M. Kahloul, S. El Arem, S. Ayachi, M. Hafsia, M. Maoua, M. Ben Othmane, O. El Maalel, W. Hmida, O. Bouallague, K. Ben Abdessalem, W. Naija & | Perceived stress and Burnout | Perceived Stress Scale version (PSS–10) and the Maslach Burnout Inventory (MBI), its French version. PSS–10 measured stress level and MBI measured burnout. The tools were | Tertiary | Primary | Quantitative results after music therapy intervention: Perceived stress scale results: Mean score decreased significantly from 22 ± 8.9 to 16 ± 7.9 ($p = 0.006$) Number of participants with | Music therapy significantly improved the stress levels of the operating theatre staff suggesting a wide use of this non-pharmacological, simple, | Individual level | |

**Table 2.** (*Continued*)

| S/N | Authors | Outcomes measured | Tools used in measurement | Facility level of workplace | Level of prevention | Quantitative results | Key findings | SEM Policy Organizational (institutions) Interpersonal (families, friends, social networks) Individual (Knowledge, attitudes, skills) | Job demand -Resources |
|---|---|---|---|---|---|---|---|---|---|
| | N. Mrizek (Kacem et al., 2020) | | administered as self-questionnaires. | | | high-stress levels decreased from 14 (41.2%) to only 4 (22.2%) Maslach burnout inventory results: Emotional exhaustion score decreased significantly from 27 ± 10.8 to 19.2 ± 9.5 ($p$ = 0.004) No significant changes in other burnout dimensions: Depersonalization ($p$ = 0.5) Professional achievement ($p$ = 0.73) Overall burnout level showed minimal change from 17.2% to 15% ($p$ = 0.98) 73.5% of participants reported that the intervention was beneficial | economical and non-invasive therapy as a preventive measure. | | |
| 11 | Kckaou, A., Dhouib, F., Kotti, N., Sallemi, I., Hammami, K. J., Masmoudi, M. L., & Hajjaji, M.(Kckaou et al., 2023) | Perceived stress, satisfaction with life and well-being | The Mindful Attention Awareness Scale (MAAS), the Perceived Stress Scale (PSS), the World Health Organisation Well-Being Index (WHO–5) and the Satisfaction with Life Scale (SWLS) as self-reporting questionnaires. | Tertiary | Primary / Secondary | Reduced perceived stress ($\beta$ = −0.30, $p$ < 0.000) and high levels of well-being ($\beta$ = 0.13, $p$ = 0.03) were associated with mindfulness | The study found that higher levels of mindfulness is significantly associated with lower perceived stress and higher levels of well-being | Individual level | |
| 12 | Frances Kelly, Margot Uys, Dana Bezuidenhout, Sarah L. Mullane and Caitlin Bristol(Kelly et al., 2021) | Knowledge, Confidence, Resilience-building behaviours, Resilience and well-being | A validated 10-item Connor-Davidson Resilience Scale (10-item CD-RISC) for Resilience, World Health Organisation–5 well-being index (WHO–5) for Well-being and other outcomes were measured using a questionairre developed for this reseach purpose. | Primary | Primary | Knowledge scores: Mean increase: 1.52 points ($p$ = 0.00) Confidence scores: Mean increase: 4.94 points ($p$ = 0.00) Resilience-building behaviors: Mean increase: 3.06 points ($p$ = 0.00) Resilience (CD-RISC): Mean increase: 3.31 points ($p$ = 0.00) Well-being (WHO–5): Mean increase: 2.58 points | Results showed significant improvements across all measured domains, suggesting that this type of e-learning intervention can be effective in supporting healthcare worker mental health during crisis periods. | Individual level | |

(*Continued*)

| S/N | Authors | Outcomes measured | Tools used in measurement | Facility level of workplace | Level of prevention | Quantitative results | Key findings | SEM Policy Organizational (institutions) Interpersonal (families, friends, social networks) Individual (Knowledge, attitudes, skills) | Job demand -Resources |
|---|---|---|---|---|---|---|---|---|---|
| | | | | | | (*p* = 0.00) Overall satisfaction rating: 4.4 out of 5 | | | |
| 13 | Idah Moyo, Livhuwani Tshivhase, & Azwihangwisi H. Mavhandu-Mudzusi(Moyo et al., 2023) | Structural outcomes include financial, human and material resources; process aspects like communication, training and support activities | | Tertiary and Secondary | Primary | The study was focused on model development using qualitative methods rather than measuring intervention outcomes quantitatively | The study identified significant gaps in the healthcare delivery system. Such as inadequate institutional support, shortages in human and material resources, and the financial burden on workers. In response, the research developed a psychosocial support model to offer guidance to support frontline workers and enhance health service delivery during COVID–19 and future public health emergencies | Individual level Organisational level | Job demand Resources |
| 14 | Şahin S, Adegbite WM, Tiryaki Şen H.(Şahin et al., 2021) | Work-to-family conflict and psychological well-being of nurses | Psychological well-being scale developed by Diener et al. (2010) Work to family conflict scale developed by Netemeyer et al. (1996), Family supportive supervisor behaviours scale developed by adaptation of scales by Clark (2001); Hammer et al. (2009); and Thompson et al. (1999) | Not indicated | Primary | Nigerian data excluded in this analysis because the data could not be obtained during Covid–19 pandemic | Family supportive supervisor behaviours contribute significantly to reduced work-to-family conflict and to better psychological well-being of nurses | Interpersonal level | |
| 15 | Enas Mahrous Abdelaziz, Iman Abdelmotelb Diab, Marwa Mohamed Ahmed Ouda, Nadia Bassiouni Elsharkawy, Fadia Ahmed Abdelkader(Abdelaziz et al., 2020) | Assertiveness skills. Psychological wellbeing and Work engagement | 1. Rathus Assertiveness Schedule; 2.Ryff's Psychological Well-Being Sclaes. 3.Utrecht Work Engagement Scale (UWES) | Tertiary | Primary | Results comparing before (T1) and after (T2) assertiveness training: Assertiveness Skills: Before (T1): Mean = 45.78 ± 11.12 After (T2): Mean = 53.75 ± 8.05 Showed statistically significant improvement (t = 4.204, *p* = .001) Psychological Well-being: | Results indicated significant improvements in assertiveness, well-being and engagement among participants. The study deemed the training feasible and potentially beneficial, recommending further research with larger samples and extended | Individual level | |

**Table 2.** (*Continued*)

| S/N | Authors | Outcomes measured | Tools used in measurement | Facility level of workplace | Level of prevention | Quantitative results | Key findings | SEM Policy Organizational (institutions) Interpersonal (families, friends, social networks) Individual (Knowledge, attitudes, skills) | Job demand -Resources |
|---|---|---|---|---|---|---|---|---|---|
| | | | | | | Before (T1): Mean = 111.3 ± 14.58 After (T2): Mean = 122.8 ± 16.46 Showed statistically significant improvement ($t$ = 4.493, $p$ = .001) Work Engagement: Before (T1): Mean = 50.08 ± 6.03 After (T2): Mean = 60.75 ± 10.72 Showed statistically significant improvement ($t$ = 5.464, $p$ = 0.001) The study also found a significant positive correlation between: Total assertiveness skills and psychological well-being scores post-intervention ($r$ = 0.431, $p$ = 0.009) | follow-up. | | |
| 16 | Mary B. Adam, Naomi Wambui Makobu, Wilson Karuri Kamiru, Simon Mbugua and Faith Mailu(Adam et al., 2021) | Personal coping mechanisms, determining staff needs, | | Tertiary | Primary | Primarily a qualitative study | Debrief sessions allowed staff members to identify their own coping strategies, learn other coping strategies from colleagues and share needs and concerns with management without fear. The rapid-cycle feedback loop facilitated management decision-making and priority-setting by allowing them to address employee issues that impacted employee well-being in real time | Individual Organisational Level | Job Demand -Resources |

(*Continued*)

**Table 2.** (Continued)

| S/N | Authors | Outcomes measured | Tools used in measurement | Facility level of workplace | Level of prevention | Quantitative results | Key findings | SEM Policy Organizational (institutions) Interpersonal (families, friends, social networks) Individual (Knowledge, attitudes, skills) Job demand-Resources |
|---|---|---|---|---|---|---|---|---|
| 17 | Wafa Abdelghaffar, Nadia Haloui, Noamen Bouchrika, Souha Yaakoubi, Amani Sarhane, Emma Kalai, Nihel Siala, Hajer Boulehmi, Souad Trabelsi, Soumaya Bourgou, Fatma Charfi, Ahlem Belhadj, Rym Rafrafi(Abdelghaffar et al., 2021) | Psychosocial Support Unit. | | Secondary | Primary/Secondary | It focus on describing implementing a psychological support unit | Despite initial stigma, the initiative received positive feedback for the support it provided through a free helpline, workshops and immediate care. It found that implementing a Psychological Support Unit (PSU) in hospitals during the pandemic could addresses mental health needs of staff and family | Individual/Organizational level |

However, two studies raised important concerns regarding the long-term sustainability of impacts from single-session trainings without follow-up support or reinforcement (Abdelaziz et al., 2020; Kckaou et al., 2023). This highlights a critical gap in the current approach to training programs and suggests a need for more longitudinal studies to assess the durability of intervention effects.

### Counselling services

Counselling services were implemented in four countries—Egypt, Kenya, Sierra Leone and Tunisia (*n* = 3 studies). These services encompassed individual and group counselling sessions (Iyamuremye and Brysiewicz, 2015; Vesel et al., 2015; Kleinau et al., 2024) and psychological helplines (Abdelghaffar et al., 2021).

Studies consistently demonstrated that counselling interventions were significantly associated with reduced stress levels among healthcare workers (Vesel et al., 2015; Abdelaziz et al., 2020; Abdelghaffar et al., 2021). For example, Waterman et al. (2018) found that their phased, CBT-based group program in Sierra Leone effectively reduced mental health symptoms among Ebola Treatment Center staff.

However, Abdelghaffar et al. (2021) noted significant barriers limiting access to these services, particularly the stigma surrounding seeking mental health support. This finding underscores the need for interventions that not only provide counselling services but also address the cultural and social barriers to accessing these services.

### Peer support programs

One study based in Sierra Leone evaluated peer support programs involving peer counselling and support groups (Vesel et al., 2015). Vesel et al. (2015) reported positive impacts on coping skills and interpersonal relationships amongst the participating healthcare workers. While these results are promising, the limited number of studies in this category highlights a need for more research into the effectiveness of peer support interventions in African healthcare settings.

### Relaxation techniques

Studies in Rwanda, Sierra Leone and Tunisia (n=3) implemented various relaxation techniques including music therapy (Kacem et al., 2020), laughter therapy (Hatzipapas et al., 2017) and physical exercise (Maingi et al., 2022). These interventions reported positive outcomes, with Kacem et al. (2020) finding lower stress and burnout risk from music therapy among operating room staff in Tunisia and Hatzipapas et al. (2017) reporting improved psychological well-being from laughter therapy among community care workers in South Africa.

Interestingly, a Tunisian study by Kckaou et al. (2023) explored associations between mindfulness and wellbeing, finding that higher mindfulness was linked to lower perceived stress and greater life satisfaction amongst healthcare workers. This aligns with broader literature on mindfulness being associated with stress and anxiety reduction, suggesting potential benefits of incorporating mindfulness-based interventions in African healthcare settings.

### Informational resources

Three studies based in Southern African countries - Botswana, Malawi and South Africa - focused on informational resources (*n* = 3) encompassing online learning courses (Kelly et al., 2021)

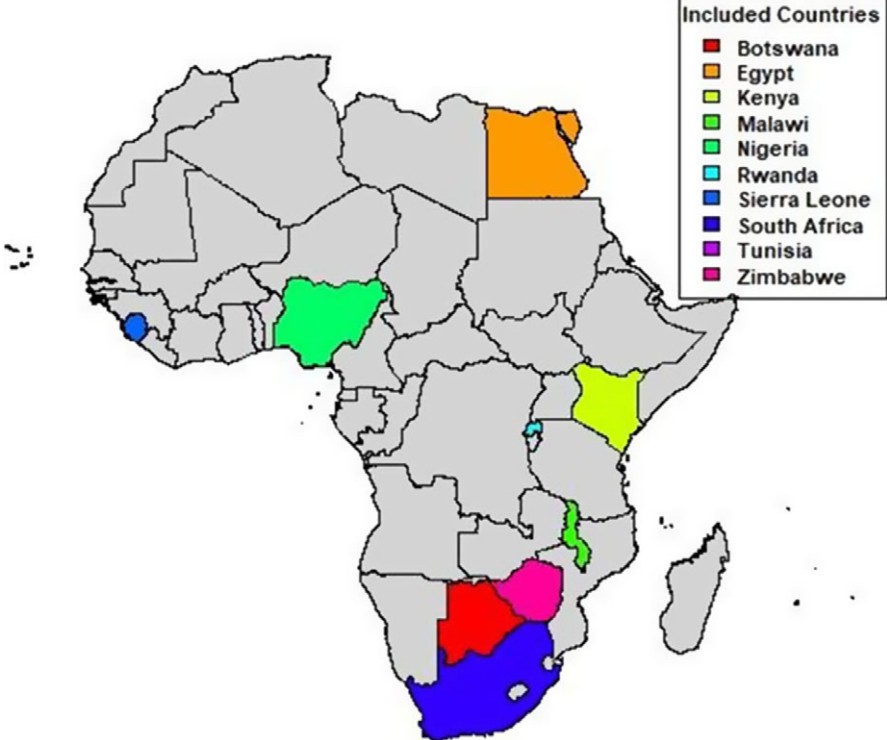

**Figure 2.** Map of countries included in the studies with intervention.

and printed guidelines/materials (Akinsulore et al., 2022; Zingela et al., 2022). These interventions were associated with increased knowledge, higher confidence, improved resilience (Kelly et al., 2021) and more effective coping skills among healthcare staff (Akinsulore et al., 2022; Zingela et al., 2022).

For instance, Kelly et al. (2021) found that their self-paced, online learning course in South Africa led to significant improvements across all measured domains of healthcare worker well-being and resilience during the COVID-19 pandemic. This suggests that digital interventions could be a promising approach, particularly in contexts where in-person interventions may be challenging to implement

### Analysis using social ecological model

#### Policy level

Only studies from Botswana indicated policy-level interventions (Ledikwe et al., 2014). Ledikwe et al. (2014) evaluated the national policy of implementing Botswana's National Workplace Wellness Program (WWP) for healthcare workers across 27 districts. The program targets both physical and mental health, revealing that physical health screenings and promotional activities are more widely adopted than occupational health and psychosocial services. Successful implementation relied on dedicated administrative support and integrating policy activities into the organizational culture, while barriers to implementation included competing work priorities, limited technical capacity for mental health services, stigma and confidentiality concerns.

#### Organizational level

Studies from Rwanda, Zimbabwe, Kenya and Tunisia indicate organisation-level interventions (Iyamuremye and Brysiewicz, 2015;

Abdelghaffar et al., 2021; Adam et al., 2021; Moyo et al., 2023). Iyamuremye and Brysiewicz (2015) in a study conducted in Rwanda demonstrated a model for managing Secondary Traumatic Stress (STS) among mental health workers through improved staffing, resources, and tools, along with organizational assessments of STS and structured protocols. The Study in Zimbabwe also demonstrated a psychosocially supportive work environment by addressing resource deficiencies and high healthcare costs, allocating adequate financial and human resources and establishing organizational counselling and communication system (Moyo et al., 2023). Abdelghaffar et al. (2021) highlighted establishment of a Psychological Support Unit (PSU) and a committee to ensure implementation while (Adam et al., 2021) demonstrated a staff well-being initiative involving debriefing sessions to share coping strategies and providing management feedback for organizational adjustments.

#### Interpersonal and individual level

Studies from Sierra Leone, Kenya and Nigeria demonstrated interpersonal-level intervention (Vesel et al., 2015; Şahin et al., 2021; Maingi et al., 2022). Specifically, Vesel et al. (2015) focused on communication skills and social connectedness between colleagues and their clients to reduce stress. Maingi et al. (2022) focused on maintaining connections with family and trusted friends to reduce fear, isolation and anxiety during health emergencies. Şahin et al. (2021) explored the impact of interpersonal levels from both colleagues and family in the form of family-supportive supervisor behaviours (FSSB) from workplace and family conflict, psychological well-being and thriving, especially during the COVID-19 pandemic. It is worth noting that 16 studies included in the review had at least one form of individual-level intervention except the study by Şahin et al. (2021) focused only on interpersonal level intervention.

### Analysis using the job demands-resources (JD-R) model's

The Job demands demonstrated by Iyamuremye and Brysiewicz (2015) included client trauma exposure to the 1994 genocide in Rwandaa and high workloads, while Adam et al. (2021) and Moyo et al. (2023) in Zimbabwe and Kenya respectively also reported high workloads, staff shortages and emotional strains. Moyo et al. (2023) highlighted insufficient protective equipment and the patient death burden due to Covid-19 while Adam et al. (2021) indicated financial strain and COVID-19-related fear. The job resourced indicated by Iyamuremye and Brysiewicz (2015) include staff education and therapeutic interventions on secondary traumatic stress (STS). Moyo et al. (2023) showed a psychosocial support model for employees and Adam et al. (2021) indicated resources that include real-time feedback, work schedule adjustments and training. These studies demonstrate the importance of balancing demands and resources to enhance employees' mental well-being and underscore the effectiveness of the JD-R model in addressing occupational challenges through context-specific resource allocation

## Discussion

This scoping review provides a comprehensive overview of workplace mental health promotion interventions for healthcare professionals in Africa. The findings reveal a diverse range of approaches being implemented across the continent, albeit with significant variations in distribution, scale and focus. This discussion will critically examine the key themes that emerged from our analysis, contextualize them within the broader literature and theoretical frameworks, and explore their implications for practice, policy, and future research.

### Diversity and distribution of interventions

Our review identified interventions from 10 African countries namely, South Africa, Botswana, Egypt, Malawi, Seirra Leon, Tunisia, Nigeria, Kenya, Rwanda and Zimbabwe. A similar review on mental health workplace intervention in Africa conducted by Hoosain et al. (2023) identified interventions only from 3 African countries (South Africa, Kenya and Botswana). The review by Hoosain et al. (2023) was not limited to health workers. Notably, this review shows an increase in countries within the continent conducting interventions on mental health intervention at the workplace. This increase in the number of countries might be as a result of the impact of the COVID-19 pandemic on health workers. This is because most of the interventions in our review aimed to mitigate the mental health impact of the pandemic on health workers. Nonetheless, the limited and uneven distribution of mental health intervention for workers in Africa likely reflects disparities in research capacity, funding and prioritization of mental health issues across different African nations. The diversity of interventions ranged from individual-level approaches such as training programs, counselling services, and relaxation techniques such as mindfulness, to interpersonal interventions such family supportive intervention, communication and social connectiveness skills to organizational-level initiatives like workplace wellness programs and creating supportive work environment through the provision of work resources to improve employee wellbeing. These interventions are like individual interventions obtained from countries in the global north to improve the mental of health workers (Shiri et al., 2023).

### Predominance of individual-level interventions

The majority of interventions identified in this review focused on individual-level approaches, particularly training programs and counseling services. This aligns with the individual level of the Social Ecological Model (SEM) (Bronfenbrenner, 1979). While these interventions showed promising short-term outcomes, their long-term effectiveness and sustainability remain weak, as highlighted by studies like Abdelaziz et al. (2020), Kckaou et al. (2023) and Shiri et al. (2023).

The emphasis on individual-level interventions may reflect the influence of Western psychological approaches and the relative ease of implementing such programs., The focus on individual-level intervention overlooks the critical role of interpersonal, organizational and systemic factors in shaping mental health outcomes. The Studies in Sierra Leone, Kenya and Nigeria highlighted the significance of social connectedness, family-supportive supervisor behaviours and maintaining connections with family and friends for mitigating stress, fear, isolation, anxiety and workplace-family conflict during health emergencies, hence showing evidence of intervention at the interpersonal level of SEM (Vesel et al., 2015; Şahin et al., 2021; Maingi et al., 2022). Evidence from Curtin et al. (2022) from a systematic review of 121 qualitative studies across 34 countries during several public health emergencies also underscored the importance of interpersonal interventions, with peer support, team cohesion and family connections fostering resilience by providing emotional and practical support.

### Limited organizational and policy-level interventions

The Job Demands-Resources (JD-R) model addresses both job demands and resources at an organizational level (Demerouti et al., 2001) which could potentially yield more sustainable improvements in mental health intervention for health workers at their work place. However, the scarcity of organizational and policy-level interventions identified in our review is a significant finding that contrasts sharply with the wealth of literature emphasizing the critical role of systemic factors in shaping workplace mental health outcomes (LaMontagne et al., 2014; Memish et al., 2017). Out of the 17 studies reviewed, only 4 (Ledikwe et al., 2017; Abdelghaffar et al., 2021; Adam et al., 2021; Moyo et al., 2023) (Adams et al., 2017; Ledikwe et al., 2017; Abdelghaffar et al., 2021; Moyo et al., 2023) explicitly addressed interventions at these macro levels. This gap can be contextualized within broader theoretical frameworks such as the Social Ecological Model (SEM) (Bronfenbrenner, 1979) and the Job Demands-Resources (JD-R) model (Demerouti et al., 2001), which underscore the importance of the organizational resource to balance the job demands of health workers. Regardless of limited evidence from our review indicating interventions at the organizational level, it reveals that organizational-level interventions, such as a supportive work environment with adequate work equipment, financial and human resources and incorporating psychosocial intervention into the organizational iterative process, are beneficial. Similarly, a comparable review by Shiri et al. (2023) focusing on countries in the global north also advocates for this degree of intervention. Shiri et al. (2023) revealed similar barriers to the ones identified from this review in engaging in organizational workplace intervention, Such barriers include insufficient personnel, excessive workloads, time constraints and the scheduling of intervention outside of working hours. The limited focus on these interventions in African healthcare settings contrasts

research from high-income countries, where organizational-level interventions have shown effectiveness in reducing occupational stress among healthcare workers (Ruotsalainen et al., 2015). This dearth of organizational and policy-level interventions may be attributed to various factors, including resource constraints, complex bureaucratic structures and the perceived immediacy of individual-level interventions. However, the few studies that did address organizational levels show promising results, aligning with emerging research on creating "psychologically healthy workplaces (Grawitch et al., 2006).

### Cultural adaptation and contextual relevance

The review revealed a concerning lack of explicit discussion around cultural adaptation of interventions. Given the diverse cultural contexts across Africa, the effectiveness of interventions likely depends heavily on their cultural appropriateness and relevance. The Cultural Adaptation Framework (Bernal et al., 2009) emphasizes the importance of adapting interventions to local contexts, considering elements such as language, metaphors and cultural concepts of mental health (Bernal et al., 2009).

Future interventions and research should prioritize cultural adaptation, ensuring that mental health promotion strategies resonate with local understandings of wellbeing and align with healthcare workers' lived experiences in different African contexts.

### Emerging innovative approaches

Despite the predominance of traditional approaches, our review identified some innovative interventions that show promise. For instance, the use of digital technologies, such as the chatbot intervention in Malawi (Kleinau et al., 2024) and the incorporation of indigenous healing practices like laughter therapy in South Africa (Hatzipapas et al., 2017), demonstrate creative ways of addressing mental health needs in resource-constrained settings. Specifically, Kleinau et al. indicates the use of chat bots by a healthy workforce for primary and secondary prevention and serves as a source for mental health information. These approaches align with global trends in digital mental health interventions (Torous et al., 2019) and the growing recognition of traditional healing practices in mental health care (Uwakwe and Otakpor, 2014). However, more research is needed to establish the long-term effectiveness and scalability of these innovative approaches across different African healthcare contexts. Future studies should consider how these interventions can be integrated into existing healthcare systems and adapted to various cultural contexts.

### Addressing stigma and barriers to access

Several studies in our review, notably Abdelghaffar et al., highlighted stigma as a significant barrier to accessing mental health support. This finding aligns with the broader literature on mental health stigma in African contexts (Kapungwe et al., 2010; Egbe et al., 2014) and underscores the need for interventions that not only provide services but also work to destigmatize mental health issues within healthcare settings. Future interventions should consider incorporating anti-stigma components and exploring ways to normalize help-seeking behaviours among healthcare professionals. This could involve awareness campaigns, open dialogues about mental health in the workplace and leadership modelling of supportive behaviours.

### Implications for practice, policy and future research

The findings of this scoping review have significant implications for practice, policy and future research in the realm of workplace mental health promotion for healthcare professionals in Africa. In practice, healthcare organizations should prioritize the implementation of multi-level interventions that address both individua, interpersonal and organizational factors, with a strong emphasis on cultural adaptation to ensure relevance and effectiveness. Policymakers need to develop and enforce regulations that support the creation and implementation of workplace mental health interventions in African healthcare settings, including allocating resources for mental health promotion and fostering supportive organizational cultures. Future research should focus on conducting longitudinal studies to assess the long-term impacts of interventions, investigating the effectiveness of organizational and policy-level approaches, exploring the process and impact of cultural adaptation, examining the cost-effectiveness and scalability of different intervention types and investigating integrated approaches that combine multiple intervention strategies. Additionally, there is a pressing need for studies that address the unique contextual factors of African healthcare settings, including resource constraints, cultural diversity and the impact of broader societal challenges on healthcare workers' mental health. By addressing these areas, researchers can contribute to the development of more effective, sustainable and culturally appropriate mental health promotion strategies for healthcare workers across Africa.

### Conclusion

This scoping review has provided a comprehensive overview of workplace mental health promotion interventions for healthcare professionals in Africa, revealing both promising approaches and significant gaps in current research and practice. While individual-level interventions, such as training programs and counselling services, have shown potential for short-term improvements, there is a critical need for more comprehensive, culturally adapted and sustainable approaches that address multiple levels of influence, as suggested by the Social Ecological Model and Job Demands-Resources model. The review has highlighted the scarcity of interpersonal, organizational and policy-level interventions, the limited attention to cultural adaptation and the emerging potential of innovative approaches such as digital interventions. Moving forward, a multi-faceted approach that integrates primary, secondary and tertiary prevention strategies, tailored to the unique cultural and resource contexts of African healthcare settings, is essential. There is also a need for more rigorous experimental, longitudinal and implementation research studies to generate high-quality evidence about intervention effectiveness and long-term impact By addressing the identified barriers, leveraging enablers and pursuing the proposed research directions, we can work towards developing more effective, sustainable and contextually appropriate mental health promotion interventions for healthcare workers across Africa. This not only has the potential to improve the wellbeing of individual healthcare professionals but also to enhance the overall quality and resilience of healthcare systems across the continent, ultimately leading to better health outcomes for both healthcare workers and the populations they serve.

**Open peer review.** To view the open peer review materials for this article, please visit http://doi.org/10.1017/gmh.2025.19.

**Supplementary material.** The supplementary material for this article can be found at http://doi.org/10.1017/gmh.2025.19.

**Data availability statement.** The data that supported the findings are available on the databases used for literature search in the study.

**Author contribution.** B.I.H. developed the protocol, Screened the articles and developed the first draft of the paper, I.O. Screened the articles, developed the tables, A.O. Screened the articles, A.Y. Screened the articles, J.Y.K. conducted database search, J.F. Screened the articles and developed figures, O.T.M. Screened article, E.E. Screened the articles and gave guidance on thematic analysis and theoretical framework. All the authors reviewed the final draft.

**Financial support.** The study was not funded by any organization.

**Competing interest.** The authors declare none.

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
