## [Reviewer Report]

This work is relevant in light of the rising burden of mental health in the global south against the backdrop of limited resources.

The manuscript has highlighted, innovative and cost-effective strategies to improve mental health. It lays a good foundation for more research into how these strategies can be strengthened or how other interventions to improve mental health outcomes at organizational and policy levels can be developed.

Including studies from several countries across the region, makes the work relevant and generalizable to the African context.

The methodological rigor employed is commendable.

---

## [Reviewer Report]

□ The scoping review contributes to the body of knowledge on mental health interventions for healthcare workers. The findings of this study are unique and crucial; in that they provide mental health promotion interventions and demonstrate the critical role mental health plays in the workplace for African healthcare workforce.

□ The document is written in a language that conforms to academic writing. The language utilised is accessible and easy to follow

□ The scoping review while it looks at the sub-Saharan Africa, meets the global criteria for reviews.

o The literature cited is drawn across the globe

o The language used is scientific and appropriate

o The scoping review is theoretically grounded, and the theoretical frameworks guided the analysis of data.

□ The results are well described and categorised into themes.

□ The discussion is well articulated. The review is highlighting pertinent issues for example the scarcity of organizational and policy-level mental health interventions which African governments, health ministries, institutions and policy makers should take note of and strive to address.

---

## [Reviewer Report]

The paper “Addressing the Mental Health Needs of Healthcare Professionals in Africa: A Scoping Review of Workplace Interventions” summarizes key findings from 17 studies in Africa based on peer reviewed and grey literature. The scoping review follows the steps outlined in the PRISMA-ScR checklist. The authors propose two conceptual frameworks to guide the review. Key information for each included study is presented in tabular form. Results are summarized by six key themes according to the type of mental health intervention. The discussion is organized by a different set of themes, partly related to the conceptual frameworks and others emerging from the studies reviewed. The conclusions recommend multi-tiered approaches for improving workplace mental health. The need for high-quality research to ensure that interventions are evidence based is not mentioned.

Overall, this scoping review is a commendable effort to identify current interventions and important gaps. Consistent with this type of review, it is qualitative in nature providing key insights into prevailing workplace mental health interventions. I recommend publication with revisions addressing three major and a few minor issues as follows.

Major issues

1. The manuscript should clearly list the mental health outcomes used by each of the 17 studies reviewed. These outcomes may include measurements of stress, burnout, anxiety, depression, resilience, satisfaction, or absenteeism among others. To judge the quality of the evidence it is important to understand whether mental health outcomes were clearly defined and quantified using standardized and validated instruments such as PHQ-9 or GAD-7. Although not a meta-analysis, the paper should cite the sample size and quantitative results for key outcomes where available or how outcomes were assessed if they were not quantified. It would also help identify the study population whether it was the general, healthy health workforce or groups with diagnosed mental health issues for which interventions and outcomes may be quite different. Without this information it will be difficult for the reader to assess whether evidence was quantified, for example, by obtaining pre-post intervention results, or was anecdotal in nature, for example, by relying exclusively on focus group discussions. Even though Table 1 mentions “Study Design”, the description of “quantitative” or “qualitative” is not informative and should be more specific.

2. Two conceptual frameworks are proposed for the review, the Social Ecological Model (SEM) and the Job Demands-Resources (JD-R) model. However, the manuscript make little use of these frameworks in the analysis and discussion. I suggest at minimum including the SEM levels and demand/resource categories in Table 1 to indicate whether the papers reviewed address them. While the discussion mentions the JD-R model once the paper lacks specifics about how either workplace demands or resources were addressed in the literature, both are important factors in mental health outcomes in the African context.

3. It is important to build on a similar previously published study “Workplace-Based Interventions for Mental Health in Africa: A Scoping Review” by Hoosain et al., 2023, and explain what value this manuscript adds, for example, studies previously not reviewed, use of conceptual frameworks, etc. The discussion should include specific references to global evidence, for example, “Effectiveness of Workplace Interventions to Improve Health and Well-Being of Health and Social Service Workers: A Narrative Review of Randomised Controlled Trials” by Shiri et al., 2023. This would help compare evidence from the African context to evidence from a high-income country context and describe how the results fit in with global research and global learnings. The conclusions should not only recommend workplace mental health interventions but also stress the need for more rigorous implementation research including experimental designs and longitudinal studies to generate high-quality evidence about intervention effectiveness and long-term impact. References to global research can help inform these recommendations.

Minor issues

4. Context matters for citations, else their use could be misleading. For example, the authors write on page 3, lines 52-55 “The workplace of healthcare workers in Africa is characterized by considerable levels of stress. This stress primarily emanates from factors such as extended working hours, heavy workloads, and limited resources (Søvold et al. 2021b).” Søvold looked at the mental wellbeing of healthcare workers in the context of COVID-19 and from a global perspective providing hardly any evidence about Africa specifically. I would have found it more appropriate to support the first sentence with publications from Africa specifically, while not a lot, a few such studies exist.

5. I suggest a more vertical format for Table 1. The horizontal layout is hard to read and has a lot of unused space.

6. Here is additional information about the article by Kleinau et al. The use of a chatbot by a healthy workforce addresses primary and secondary prevention. The chatbot also serves as a source for mental health information and exercises. Perhaps the authors can reflect this in their manuscript.

---

## [Editor Report]

Thank you for the manuscript on a very important topic.

Some of the workplace interventions identified are from the COVID-19 pandemic. It will be worth including a paragraph in the discussion section about this.

---

## [Reviewer Report]

A wholehearted thank you to the authors for addressing comments to the first draft of the paper. The application of the two theoretical frameworks is much improved. The information added made this paper much more informative about workplace mental health promotion for healthcare professionals in Africa.

While the added content is great, the added text should be thoroughly copy edited especially throughout the manuscript. Tables 1 and 2 are still hard to read in their current layout. They can be combined and presented in a vertical format.

One instant needs clarification. On Page 39, lines 292-294, please check grammar. Should it say, “It is worth noting that 16 studies included in the review had at least one form of individual-level interventions except the study by Şahin et al., (2021)Şahin et al., (2021), which focused only on interpersonal level interventions.”